# An Insight into Role of Auditory Brainstem in Tinnitus: A Systematic Review of Diagnostic Assessments

**DOI:** 10.3390/audiolres15060149

**Published:** 2025-11-06

**Authors:** Giovanni Freda, Andrea Ciorba, Nicola Serra, Rita Malesci, Francesco Stomeo, Chiara Bianchini, Stefano Pelucchi, Pasqualina Maria Picciotti, Luigi Maiolino, Giacinto Asprella Libonati, Anna Rita Fetoni

**Affiliations:** 1Department of Neuroscience, Reproductive Sciences and Dentistry, Audiology Section University of Naples Federico II, Via Pansini 5, 80131 Naples, Italy; dott.giovannifreda@gmail.com (G.F.); rita.malesci@unina.it (R.M.); 2ENT and Audiology Unit, Department of Neuroscience and Rehabilitation, University Hospital of Ferrara, 44121 Ferrara, Italy; andrea.ciorba@unife.it (A.C.); stmfnc@unife.it (F.S.); chiara.bianchini@unife.it (C.B.); stefano.pelucchi@unife.it (S.P.); 3Hearing and Balance Unit, Department of Head and Neck, Federico II University Hospital, Via Pansini 5, 80131 Naples, Italy; 4Complex Operational Unit of Ear Nose and Throat, Department of Neuroscience, Sense Organs and Thorax, Foundation Polyclinic University A. Gemelli IRCCS, 00168 Rome, Italy; pasqualinamaria.picciotti@unicatt.it; 5Department of Neuroscience, Sense Organs and Thorax, Catholic University of the Sacred Heart, 00168 Rome, Italy; 6Department of Medical, Surgical Sciences and Advanced Technologies G.F. Ingrassia, University of Catania, 95123 Catania, Italy; luigi.maiolino@unict.it; 7Department of Otorhinolaryngology and Cervico-Facial Surgery, Madonna delle Grazie Hospital, 75100 Matera, Italy; asprella@tin.it

**Keywords:** tinnitus, brainstem, diagnosis, auditory brainstem response

## Abstract

**Background/Objectives**: Tinnitus is a complex auditory phenomenon with multifactorial origins, often involving both peripheral and central auditory pathways. Given the multifactorial nature of tinnitus, this review specifically focuses on the auditory brainstem as it represents the first central relay for auditory input and a key site of abnormal synchrony and central gain, which may generate or modulate tinnitus even when peripheral hearing appears normal. Several studies suggest a potential role of brainstem dysfunction in its pathogenesis, even among patients with normal hearing thresholds. Although the physiopathological data provide evidence for the role of brainstem in the generation and magnification of tinnitus, the diagnostic tools are still unclear. This systematic review aimed to investigate the diagnostic relevance of brainstem-level abnormalities in individuals with tinnitus. **Methods**: Following PRISMA guidelines, a literature search was conducted using PubMed, Scopus, and Web of Science from January 2000 to June 2025. Studies were included if they addressed the diagnostic relationship between tinnitus and brainstem involvement. Data on auditory brainstem response (ABR), otoacoustic emissions (used to differentiate peripheral from central auditory abnormalities), neuroimaging, and electrophysiological markers were extracted. **Results**: Twenty studies were included. Most used ABR as a diagnostic tool, revealing significant amplitude and latency alterations in tinnitus patients compared to controls, particularly in wave V and V/I amplitude ratios. Imaging studies supported altered brainstem–cortical connectivity and localized changes in inferior colliculus (IC) activity. Additional techniques, such as middle-latency evoked potentials and gap-in-noise detection, showed potential but lacked consistent clinical utility. **Conclusions**: Evidence suggests that brainstem dysfunction may contribute to tinnitus generation or persistence. ABR and advanced imaging represent specific diagnostic tools, though standardization and high-quality studies are still needed to improve clinical applicability.

## 1. Introduction

Tinnitus is a common and often debilitating auditory symptom characterized by the perception of sound in the absence of an external stimulus [1]. Tinnitus can be classified as objective (rare, with an identifiable physical sound source) or subjective (the vast majority). Subjective tinnitus is further categorized as primary (idiopathic, without any identifiable cause) or secondary (resulting from identifiable causes such as vestibular schwannoma, Ménière’s disease, or other otologic/neurologic disorders). It affects approximately 10–15% of the adult population and is associated with a variety of auditory and non-auditory comorbidities, including hearing loss, anxiety, depression, and sleep disturbances [1,2,3]. Despite its prevalence, the pathophysiology of tinnitus remains incompletely understood, particularly in individuals with normal hearing thresholds [4]. Increasing evidence suggests that tinnitus is not solely a peripheral phenomenon, but rather a complex auditory-perceptual disorder involving central nervous system structures, especially the brainstem [5].

The brainstem, as a central relay station in the auditory pathway, plays a crucial role in sound processing and auditory signal transmission. Structures such as the cochlear nucleus (CN), superior olivary complex, lateral lemniscus, and inferior colliculus (IC) are involved in the early integration and modulation of acoustic input. Dysfunction at the level of these nuclei has been linked to altered neural synchrony, increased central gain, and maladaptive plasticity, all of which are proposed mechanisms in the genesis and maintenance of tinnitus [6]. In this context, brainstem dysfunction appears to contribute not only to the generation of tinnitus but also to the modulation and aggravation of tinnitus-related symptoms. Indeed, alterations in brainstem nuclei and their interconnections with the limbic and prefrontal systems may influence the emotional and cognitive response to the phantom sound. Recent studies demonstrated that chronic tinnitus in older adults is often associated with anxiety, depressive symptoms, and impaired cognitive control, which may reflect abnormal connectivity between auditory and non-auditory brainstem pathways and higher cortical centers involved in attention and emotional regulation [2,3]. These findings support the hypothesis that the brainstem acts as an interface between auditory perception and emotional–cognitive processing, thereby amplifying tinnitus distress and its impact on quality of life.

Animal studies suggest that brainstem dysfunctions include increased firing rates (hyperactivity), bursting activity, and abnormal neural synchrony in the CN—particularly in the dorsal CN—and in the IC [7,8]. Interestingly, in patients with gaze-evoked tinnitus following the removal of a vestibulocochlear (VIII) nerve schwannoma, an increase in tinnitus loudness has been observed, associated with heightened activity in the CN and IC and reduced inhibition in the auditory cortex. This is consistent with plastic reorganization in the brainstem after surgery [9].

The modulatory activity of the brainstem is also important because it receives input from various auditory and non-auditory pathways, which can influence the initial tinnitus signal or its perception. This may explain several tinnitus-related phenomena, such as the effects of contralateral masking of tinnitus with noise [10], fluctuations in tinnitus loudness and pitch [11], and residual inhibition (RI) or transient suppression of tinnitus following sound exposure [12,13,14]. Thus, efferent pathways—including the olivocochlear, colliculo-cochlear, and lemnisco-cochlear pathways—appear to be mainly implicated in the modulation of hyperactivity (increased spontaneous firing of central auditory neurons observed after peripheral auditory deafferentation) associated with tinnitus perception [15,16].

In non-auditory–induced tinnitus, brainstem modulation is also relevant in several contexts, including somatic tinnitus originating from the connection between afferent cranial nerves and the trigeminal nerve and nucleus toward the CN and IC; attention-induced changes involving the IC and pontomesencephalic tegmentum [17]; and emotional or mood variations [18]. Notably, the severity of tinnitus has been associated with stress, anxiety, and depression, regardless of the patient’s age [3,18,19]. Depressive symptoms can play a key role in inducing chronic tinnitus in a sort of “vicious cycle” [20], which is also relevant from a therapeutic perspective. The major pathways involved in this cycle include midbrain structures such as the locus coeruleus—which modulates anxiety and sleep disorders—and the dorsal raphe nucleus, which plays a role in the brain’s serotonin system and its connections to the auditory pathways. This has been recently confirmed by a postmortem study showing a reduced number of serotonergic cell bodies in tinnitus patients [21].

The complexity of the pathophysiological pathways involved in both the generation and modulation of phantom sound perception [22] contributes to the diagnostic challenges of tinnitus, which remain significant even in light of therapeutic considerations.

From a diagnostic standpoint, several techniques have been proposed to assess brainstem function in tinnitus patients. These include auditory brainstem responses (ABRs), middle- and late-latency evoked potentials, otoacoustic emissions, magnetic resonance imaging (MRI), and, more recently, functional imaging techniques such as functional magnetic resonance imaging (fMRI). Otoacoustic emissions (OAEs), although primarily used to assess outer hair cell integrity, were included in this review because they help exclude peripheral dysfunction when investigating potential central auditory abnormalities. Middle- and late-latency auditory evoked potentials (AMLR and ALLR), while partly reflecting cortical processing, complement ABRs by characterizing central conduction from the brainstem to auditory cortex. These tools have been used to investigate potential neurophysiological and anatomical correlates of tinnitus [23,24]. However, their diagnostic value remains debated, partly due to methodological heterogeneity and the lack of standardized protocols.

To date, few systematic reviews have addressed the specific role of brainstem involvement in the diagnosis of tinnitus. Most published reviews focus broadly on tinnitus pathophysiology or management, without dissecting the contribution of brainstem-level findings to differential diagnosis or clinical evaluation. Given this gap, a comprehensive synthesis of studies exploring the diagnostic relationship between tinnitus and brainstem abnormalities is warranted.

This systematic review aims to critically examine the available literature on the diagnosis of brainstem-related pathologies in patients with tinnitus. In particular, it seeks to identify diagnostic markers and tools that may help detect central auditory dysfunction and clarify the potential role of the brainstem in tinnitus perception.

## 2. Materials and Methods

The systematic review was conducted following the Preferred Reporting Items for Systematic Review and Meta-Analysis (PRISMA 2020). The PRISMA statement 2020 [25].

The study protocol has been stored at the following link: “https://osf.io/yh3t7/ (accessed on 30 August 2025)”, with the following registration doi: https://doi.org/10.17605/OSF.IO/UNXYV.

The research question for this systematic review was: “What diagnostic tools can identify brainstem pathology contributing to tinnitus?”.

### 2.1. Inclusion Criteria and Eligibility

The literature review process was conducted in the following steps: identification of the research questions through the PIOS (Population, Intervention, Outcome, Design) method, literature search, included papers selection, findings appraisal, and summary building. Eligible studies should meet the following criteria: (1) population: adult population; (2) intervention: diagnostic assessment of brainstem function in tinnitus patients; (3) outcome: improve understanding of the role of brainstem pathologies related to tinnitus (4) design: observational (case–control, cohort) and randomized controlled trials (RCTs) or cohort studies.

Theses, posters, commentaries, meta-analyses, letters to the editor, reviews, books, conference papers, study protocols, technical reports and case reports were excluded. Finally, only peer-reviewed articles in English language were considered.

### 2.2. Search Strategy

To identify clinical studies on tinnitus connected to the diagnosis of brainstem pathologies, the manuscripts were searched on PubMed Central (National Center for Biotechnology Information, Bethesda, MD, USA), Web of Science, and Scopus. The search strings used, and the number of articles found were reported in Table 1. The literature search was conducted in July 2025.

### 2.3. Study Selection and Data Extraction

Two independent authors, a medical doctor resident in audiology with more than 3 years of experience and a biostatistician with more than 20 years of experience (G.F., N.S.) screened titles and abstracts according to the search strategy focused on the relationship between (tinnitus) and (brainstem) and (diagnosis). First, the authors read the titles and abstracts of the articles and selected those that were interesting while being as inclusive as possible. Following the first phase, they independently assessed the full text of all potentially relevant studies for inclusion in this review. Any disagreement was resolved through discussion with a third author (full professor in audiology with high level of experience, A.R.F.). Then, using a standardized data collection form, the following information was extracted from the included studies: first author, journal, publication year, title, database, study aim, type of study and sample size, and results.

The inclusion criteria were primary research studies (including descriptive and observational studies, RCTs, and basic science articles) published from 1 January 2000 to 30 June 2025 on the tinnitus, brainstem and diagnosis connected to brainstem pathology. We excluded those that did not fit the inclusion criteria or that directly addressed the topic under investigation; in particular, we excluded all articles that referred to therapy or focused on diseases in which the tinnitus was only partially described.

### 2.4. Quality Assessment

The Quality Assessment was performed with Effective Public Health Practice Project Quality Assessment Tool (EPHPP) [26]. The EPHPP tool was developed for systematic reviews of public health topics and can be used in different study designs. It is characterized by six components: “se-lection bias”, “study design”, “confounding factors”, “blinding”, “data collection methods”, and “withdrawals and dropouts “. The partial and global score assigned for the assessment of quality level were Weak, moderate, or strong [26].

## 3. Results

Overall, our search generated 952 articles by all databases. We removed 910 articles because they did not meet our inclusion criteria. In particular, 700 articles were excluded due to the study design (121 reviews, 9 conference papers, 9 book chapters, 260 case reports, 1 commentary, 1 study protocol, 2 meta-analysis, 7 letters, 9 preclinical studies, and 281 others, i.e., all other types of study described in PubMed), 92 were in other languages (17 Polish, 18 Chinese, 26 German, 10 Spanish, 7 Japanese, 6 French, 3 Portuguese, 2 Italian, 1 Croatian, 1 Czech, 1 Serbian) and 121 had a different topic. Of the remaining 42 articles, 7 have been removed because they were duplicates. Finally, we excluded 15 articles: 5 articles that did not involve at least one of the keywords, 9 that focused on other topics and 1 because it was a model validation. As a result, our systematic search provided 20 articles. The details of the research performed are shown in the flowchart in Figure 1.

Table 2 provides a description of all the included papers. This includes the first author, the title, the journal in which it was published, the database in which it was found, the aim of the study, the study type, the sample size, the tools used, and the results of the study.

### Quality Assessment Results

Two reviewers (R1 and R2) independently conducted the quality assessment of the articles included in our study. Both reviewers were researchers with expertise in reviewing methods and critical appraisal, rousing the EPHPP tool. Each reviewer expressed an opinion for each article. The opinions were assessed by a third reviewer represented by a full professor in audiology with more than 30 years of experience who expressed the final assessment. In Table 3, the quality of all included studies was reported.

## 4. Discussion

The extensive literature on the evidence supporting the physiopathological role of the brainstem in the generation and modulation of tinnitus has not yet led to significant improvements in the diagnostic challenge. This systematic review confirms that multiple diagnostic strategies have been employed to investigate brainstem involvement in tinnitus, particularly in patients with normal audiograms. Specifically, six main diagnostic approaches emerged from the reviewed studies, encompassing both electrophysiological and neuroimaging methods. These include ABR, ECochG, MLR, and FFR—used to explore neural conduction and synchronization along the auditory pathway—as well as MRI and fNIRS, which provide complementary structural and functional information on auditory brainstem and cortical networks. The following sections describe these tools in detail according to their methodological features and clinical relevance.

The most frequently used tools were ABRs, which consistently revealed subtle abnormalities in wave latencies and amplitudes—most notably prolonged Inter-peak Latency and altered V/I amplitude ratios in tinnitus patients [27,34]. Several studies, such as those by Mahmoudian et al. and Makar et al., highlighted significant deviations in ABR parameters between tinnitus subjects and controls, indicating a potential dysfunction at the level of the auditory brainstem—even in the absence of measurable hearing loss [27,29]. This supports the more recent hypothesis of “hidden hearing loss,” in which synaptopathy and impaired neural synchrony may not manifest in standard audiometry but can be detected through neurophysiological markers [45].

Despite ABRs being traditionally used for the evaluation of extracochlear lesions, recent insights into hidden hearing loss, tinnitus pathophysiology, and hyperacusis have renewed interest in the role of wave parameter analysis. Experimental models of noise exposure have shown that wave I amplitude reduction and increased latency are associated with synaptopathy/deafferentation [47]. However, clinically, no standardized normative values are currently available to predictably diagnose synaptopathy based on wave I modifications. According to Bramhall (2021) [48], increasing knowledge on stimulus parameters and recording levels will provide more evidence of the role of ABRs in the diagnosis of synaptic damage and VIII nerve deafferentation.

Figure 2 illustrates the anatomical course of the central auditory pathway, highlighting the neural generators of each ABR wave from the auditory nerve to the IC. This visual reference aids in contextualizing the electrophysiological alterations observed in tinnitus patients.

Interestingly, functional assessments—including temporal processing measures such as gap-in-noise detection [43] and middle-latency auditory potentials [40]—suggest that tinnitus may be associated with broader temporal and cortical processing dysfunctions. However, as previously noted, these results remain exploratory due to heterogeneity in testing protocols. A recent multivariate meta-analysis in tinnitus patients with normal hearing reported significantly longer latencies of ABR waves I (standardized mean difference = 0.66 ms, *p* < 0.001), III (standardized mean difference = 0.43 ms, *p* < 0.001), and V (standardized mean difference = 0.47 ms, *p* < 0.01). While other auditory evoked potential techniques, such as middle-latency responses and frequency-following responses, have been reported as inconclusive for tinnitus evaluation [49], ABR recording remains the benchmark for assessing brainstem alterations in tinnitus patients.

Given the well-established role of MRI in identifying retrocochlear and brainstem pathology, the integration of electrophysiological tests—particularly ABRs—into the modern diagnostic approach may offer additional value. ABR evaluation, especially in normal-hearing cases, could serve as a valuable tool for both assessing and potentially predicting the course of tinnitus, as proposed by several authors [49]. Parameters such as I–III or I–V interpeak intervals and wave amplitudes might even help predict tinnitus relapse in selected cases [50]. In the absence of a specific diagnostic tool to assess tinnitus in clinical practice, electrophysiological testing, and ABRs in particular, remain key for evaluating auditory pathway activity and identifying potential pathophysiological conditions such as synaptopathy.

High-resolution imaging, such as three-dimensional fast imaging employing steady-state acquisition (3D-FIESTA) and contrast-enhanced MRI, has also provided valuable insights. In clinical practice, contrast-enhanced MRI is widely used for evaluating the temporal bone in patients with audiovestibular dysfunction and tinnitus. The introduction of 3D-FIESTA sequences offers much higher spatial resolution, enabling clear visualization of small structures such as cranial nerves and fine anatomical details. These sequences also allow faster acquisition times, improving patient compliance and workflow efficiency [51]. Studies by Oh et al. and Schick et al. have shown that these tools can detect subtle retrocochlear anomalies, supporting their role as complementary diagnostic methods in unilateral tinnitus [30,39]. Positive findings identifiable through these techniques are summarized in Table 4.

Studies using fMRI [36,40]—arguably the most intriguing diagnostic tool—have further demonstrated altered connectivity and lateralization within central auditory networks, particularly involving the IC and auditory cortex. These findings suggest that tinnitus may result from maladaptive plasticity involving brainstem–cortical loops rather than purely peripheral lesions.

Concerning other possible markers (e.g., blood biomarkers) for brainstem dysfunction in primary tinnitus, no single, definitive diagnostic tool has yet been identified. However, in other conditions unrelated to tinnitus, the presence of specific markers—such as neurofilament light chain, correlated with brainstem and peripheral nerve damage—has been proposed [52].

The development of animal models may help elucidate the neuronal mechanisms of tinnitus generation, including the cellular processes underlying its onset [49]. Proposed models could focus on neuronal hyperactivity, synaptopathy, and neural reorganization, particularly in brainstem nuclei such as the CN, superior olivary complex, lateral lemniscus, and IC [53].

It is likely that, in the near future, additional tools will become available for evaluating primary tinnitus and, in particular, for characterizing its pathophysiological features. This could eventually enable more personalized assessments and potentially specific treatment strategies.

Future research should focus on longitudinal, well-controlled investigations integrating multimodal diagnostic tools. Developing standardized protocols for ABR interpretation and functional imaging could enhance clinical applicability and potentially support personalized diagnostic and therapeutic strategies in tinnitus management.

## 5. Limitations

This review has several limitations. First, the EPHPP evaluation indicated that most included studies were of weak or moderate methodological quality, with only about 15% rated as strong. Second, sample sizes were often small, limiting the generalizability of findings. Third, prospective investigations were scarce, with most data derived from retrospective or cross-sectional designs. Finally, the review deliberately focused on the brainstem’s involvement in tinnitus, excluding cortical and extra-auditory networks that may also contribute to its pathophysiology.

## 6. Conclusions

This review indicates that the brainstem plays a central role in the pathophysiology and potential diagnosis of tinnitus, even among patients without audiometric hearing loss. Given the well-established role of neuroimaging in detecting retrocochlear and brainstem pathology, it is now possible to reconsider the modern application of electrophysiological tests. ABR abnormalities—especially those involving wave V parameters—and alterations in brainstem–cortical connectivity represent promising diagnostic targets.

Advanced imaging techniques and electrophysiological paradigms may assist in identifying specific subtypes of tinnitus linked to central auditory dysfunction. Nevertheless, the existing evidence is limited by methodological variability and moderate-to-low study quality.

## Figures and Tables

**Figure 1 audiolres-15-00149-f001:**
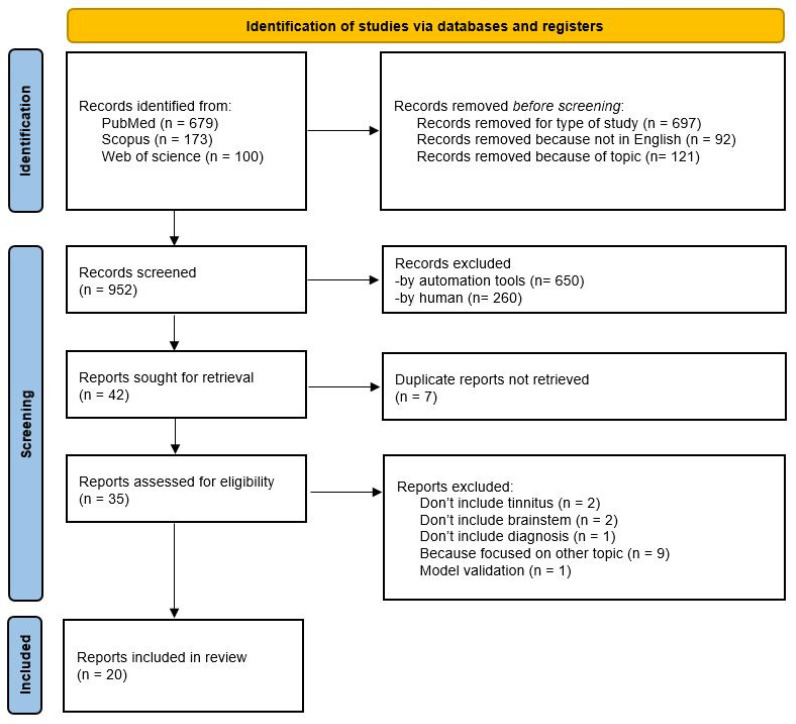
PRISMA flow diagram 2020. The flowchart displays article search and selection [25].

**Figure 2 audiolres-15-00149-f002:**
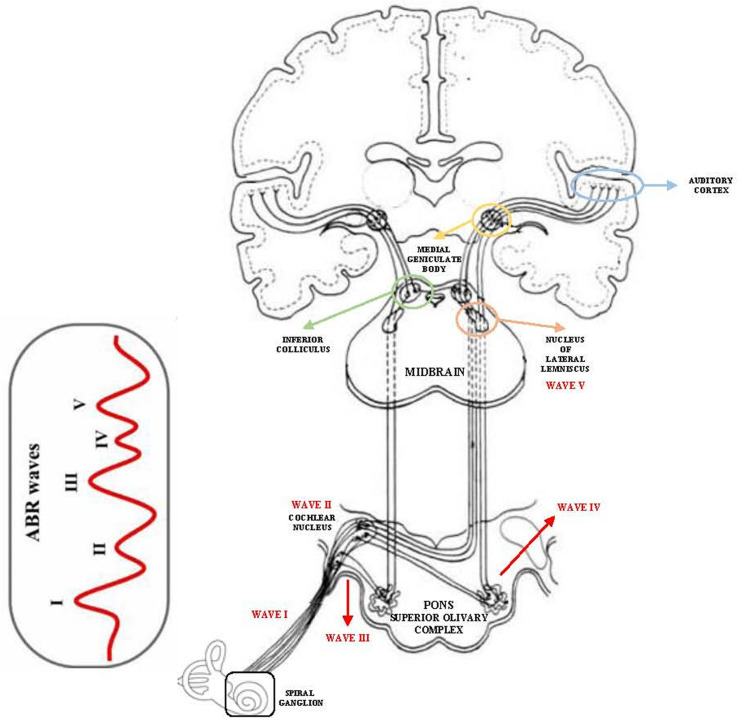
Schematic representation of the central auditory pathway and corresponding neural generators of ABR waves.

**Table 1 audiolres-15-00149-t001:** Database, search strings and number of articles found.

Database	Search Strings	Number of Results
PubMed	((“tinnitus”[MeSH Terms] OR “tinnitus”[All Fields]) AND (“brain stem”[MeSH Terms] OR (“brain”[All Fields] AND “stem”[All Fields]) OR “brain stem”[All Fields] OR “brainstem”[All Fields] OR “brainstems”[All Fields] OR “brainstem s”[All Fields]) AND (“diagnosable”[All Fields] OR “diagnosi”[All Fields] OR “diagnosis”[MeSH Terms] OR “diagnosis”[All Fields] OR “diagnose”[All Fields] OR “diagnosed”[All Fields] OR “diagnoses”[All Fields] OR “diagnosing”[All Fields] OR “diagnosis”[MeSH Subheading])) AND (2000/1/1:2025/6/30[pdat])	679
Scopus	(TITLE-ABS-KEY (tinnitus) AND TITLE-ABS-KEY (brainstem) AND TITLE-ABS-KEY (diagnosis)) AND PUBYEAR > 1999 AND PUBYEAR < 2026	173
Web of Science	Tinnitus (All Fields) and brainstem (All Fields) and diagnosis (All Fields)	100

**Table 2 audiolres-15-00149-t002:** Description of included papers.

First Author, Title (IT/EN), Journal (Year)	Database	Topic Aim of the Study	Type of Study, Sample Size, Tools	Results
(1) Saeid Mahmoudian, Alterations in early auditory evoked potentials and brainstem transmission time associated with tinnitus residual inhibition induced by auditory electrical stimulation. International Tinnitus Journal (2013) [27]	PubMed	The main aim of this study was to evaluate the effects of RI induced by auditory electrical stimulation (AES) in the primary auditory pathways using early auditory-evoked potentials in subjective idiopathic tinnitus subjects.	Type of study: “cross-sectional and randomized, placebo-controlled trial” Sample size: 44 Tools: Electrocochleography, ABR, brainstem transmission time pre/post-AES	Patients with residual inhibition showed significant changes in compound action potential amplitude and in the I/V and III/V amplitude ratios. The brainstem transmission time was reduced after stimulation in those who experienced residual inhibition.
(2) Haúla F. Haider, Audiological biomarkers of tinnitus in an older Portuguese population. Frontiers in Aging Neuroscience. (2022) [28]	Scopus	The main aim of this study was to evaluate the associations between audiological parameters and the presence or severity of tinnitus, to improve tinnitus diagnosis, treatment, and prognosis.	Type of study: not specified Sample size: 122 Tools: Audiometry, psychoacoustic evaluation, ABR, DPOAEs, Tinnitus Handicap Inventory	Exposure to noise and hearing loss were associated with a higher risk of tinnitus. An abrupt onset and moderate-to-severe hyperacusis were also related to more severe tinnitus. Increased wave I amplitude appeared protective, while a higher V/I amplitude ratio was linked to a greater likelihood of moderate tinnitus.
(3) Sujoy Kumar Makar, Auditory System Synchronization and Cochlear Function in Patients with Normal Hearing With Tinnitus: Comparison of Multiple Feature with Longer Duration and Single Feature with Shorter Duration Tinnitus. International Tinnitus Journal (2017) [29]	PubMed	To observe cochlear and brainstem function in normal hearing ears with tinnitus using DPOAEs and ABR audiometry	Type of study: Case-control study. Sample size: 60 (control group: 30; study group: 30) Tools: Pure tone audiometry, tinnitus matching, DPOAEs, ABR latencies/Inter-peak Latency	Ears with tinnitus showed a reduction of DPOAE signal-to-noise ratio and amplitude, together with prolonged ABR latencies. Abnormalities of the III–V inter-peak latency were particularly associated with longer duration and more complex tinnitus.
(4) Jae Ho Oh, Clinical Application of 3D-FIESTA Image in Patients with Unilateral Inner Ear Symptom. Korean J Audiol (2013) [30]	Scopus	The purpose of this study was to introduce the clinical usefulness of three-dimensional fast imaging employing steady-state acquisition (3D-FIESTA) MRI in patients with unilateral ear symptoms.	Type of study: not specified Sample size: 253 Tools: 3D-FIESTA MRI temporal bone	Imaging revealed a variety of abnormalities including acoustic neuroma, enlarged vestibular aqueduct syndrome, vascular aneurysms and other cochlear or nerve malformations. The technique proved useful for identifying subtle cochlear and retrocochlear pathologies.
(5) Shadman Nemati, Cochlear and Brainstem Audiologic Findings in Normal Hearing Tinnitus Subjects in Comparison with Non-Tinnitus Control Group. Acta Medica Iranica. (2014) [31]	PubMed	Present study was performed in order to better understanding of the probable causes of tinnitus and to investigate possible changes in the cochlear and auditory brainstem function in normal hearing patients with chronic tinnitus.	Type of study: cross-sectional, descriptive and analytic study Sample size: 25 (control group: 6, study group: 19) Tools: TEOAEs, ABR latencies/amplitudes	No significant differences were found in absolute wave latencies or interpeak intervals between tinnitus and control groups. Only the V/I amplitude ratio was significantly higher in tinnitus patients, suggesting a potential marker of altered brainstem function.
(6) Helga M. Kehrle, Comparison of Auditory Brainstem Response Results in Normal-Hearing Patients With and Without Tinnitus. Arch Otolaryngol Head Neck Surg (2008) [32]	PubMed	To perform an electrophysiological evaluation of the auditory nerve and the auditory brainstem function of patients with tinnitus and normal-hearing thresholds using the ABR.	Type of study: case-control study Sample size: 75 (control group: 38, study group: 37) Tools: ABR latency & interpeak measures	Almost half of the tinnitus patients showed at least one abnormal ABR parameter. Waves I, III and V were significantly delayed compared with controls, and the V/I amplitude ratio was higher in tinnitus despite being within normal limits.
(7) Olaf Zagólski, Comparison of characteristics observed in tinnitus patients with unilateral vs. bilateral symptoms, with both normal hearing threshold and distortion-product otoacoustic emissions. Acta Oto-Laryngologica (2016) [33]	PubMed	The study was to determine the differences between tinnitus characteristics observed in patients with unilateral vs. bilateral symptoms and normal hearing threshold, as well as normal results of DPOAEs.	Type of study: not specified Sample size: 380 Tools: Tympanometry, ABR, psychoacoustic tests	Bilateral tinnitus was more often associated with longer symptom duration, greater sound hypersensitivity and higher pitch compared with unilateral cases. Noise exposure was the most frequent reported trigger in bilateral tinnitus.
(8) Hyun Joon Shim, Comparisons of auditory brainstem response and sound level tolerance in tinnitus ears and non-tinnitus ears in unilateral tinnitus patients with normal audiograms. PLoS ONE (2017) [34]	PubMed	Recently, “hidden hearing loss” with cochlear synaptopathy has been suggested as a potential pathophysiology of tinnitus in individuals with a normal hearing threshold. Several studies have demonstrated that subjects with tinnitus and normal audiograms show significantly reduced ABR wave I amplitudes compared with control sub- jects, but normal wave V amplitudes, suggesting increased central auditory gain. We aimed to reconfirm the “hidden hearing loss” theory through a within-subject comparison of wave I and wave V amplitudes and UCL, which might be decreased with increased central gain, in TEs and non-TEs.	Type of study: not specified Sample size: 61 (control group: 18, study group: 43) Tools: ABR wave I & V, UCL tests	The study did not find conclusive ABR evidence of cochlear synaptopathy. However, patients with tinnitus showed reduced sound tolerance, which may indicate increased central gain even with normal audiograms.
(9) Aysun Coskunoglu, Evidence of associations between brain-derived neurotrophic factor (BDNF) serum levels and gene polymorphisms with tinnitus. Noise Health (2017) [35]	Scopus	This study aims to investigate whether there is any role of BDNF changes in the pathophysiology of tinnitus.	Type of study: not specified Sample size: 94 (control group: 42, study group: 65) Tools: Tinnitus Handicap Inventory, psychiatric interview, BDNF gene & serum analysis	Patients with tinnitus had lower serum BDNF levels compared with controls. No significant association was found between BDNF gene polymorphisms and tinnitus, suggesting that reduced BDNF might play a role in its pathophysiology.
(10) C.P. Lanting, Functional imaging of unilateral tinnitus using fMRI. Acta Oto-Laryngologica (2008) [36]	PubMed	The major aim of this study was to determine tinnitus-related neural activity in the central auditory system of unilateral tinnitus subjects and compare this to control subjects without tinnitus.	Type of study: not specified Sample size: 22 (control group: 12, study group: 10) Tools: fMRI (IC & auditory cortex)	Functional imaging showed a prominent role of the inferior colliculus in tinnitus-related neural activity, highlighting its importance in central auditory processing of tinnitus.
(11) Hsiang-Hung Lee, Impact of tinnitus on chirp-evoked auditory brainstem response recorded using maximum length sequences. Acoustical Society of America (2025) [37]	Scopus	The etiopathogenesis in audiovestibular symptoms can be elusive, despite extensive differential diagnosis. This article addresses the value of MRI in analysis of the complete audiovestibular pathway.	Type of study: retrospective Sample size: 40 (control group: 20, study group: 20) Tools: Chirp ABR amplitudes/latencies at various rates	Tinnitus patients demonstrated larger wave I amplitudes, prolonged wave V latencies and longer interpeak intervals compared with controls, indicating altered brainstem auditory processing.
(12) Eui-Cheol Nam, Is it necessary to differentiate tinnitus from auditory hallucination in schizophrenic patients? The Journal of Laryngology & Otology (2005) [38]	Scopus	This study examined whether the differentiation of tinnitus from auditory hallucination is necessary for the proper management of these symptoms in schizophrenic patients.	Type of study: not specified Sample size: 15 Tools: Pure tone audiometry, ABR	Auditory brainstem responses were abnormal only in patients with pure auditory hallucinations, supporting the need to distinguish tinnitus from hallucinations in the clinical evaluation of schizophrenia.
(13) Bernhard Schick, Magnetic Resonance Imaging in Patients with Sudden Hearing Loss, Tinnitus and Vertigo. Otology & Neurotology (2001) [39]	Scopus	The etiopathogenesis in audiovestibular symptoms can be elusive, despite extensive differential diagnosis. This article addresses the value of MRI in analysis of the complete audiovestibular pathway.	Type of study: retrospective Sample size: 354 Tools: Contrast-enhanced MRI	Magnetic resonance imaging detected a variety of abnormalities including microangiopathic brain changes, cerebellopontine angle tumors, demyelinating lesions and inflammatory processes, confirming its high diagnostic value in audiovestibular disorders.
(14) Valdete Alves Valentins dos Santos Filha, Middle Latency Auditory Evoked Potential (MLAEP) in Workers with and without Tinnitus who are Exposed to Occupational Noise. Med Sci Monit (2015) [40]	PubMed	Tinnitus is an important occupational health concern, but few studies have focused on the central auditory pathways of workers with a history of occupational noise exposure. Thus, we analyzed the central auditory pathways of workers with a history of occupational noise exposure who had normal hearing threshold, and compared MLAEP in those with and without noise-induced tinnitus.	Type of study: cross-sectional Sample size: 60 (control group: 30, study group: 30) Tools: Audiometry, MLAEP	Both tinnitus and non-tinnitus groups had prolonged MLAEP latencies, but tinnitus patients tended to show more amplitude alterations, suggesting subtle central auditory processing impairment due to occupational noise.
(15) Claus-F. Claussen, Neurootological Differentiations in Endogenous Tinnitus. International Tinnitus Journal (2009) [41]	Scopus	Vertigo and tinnitus are very frequent complaints. Often, we find multisensory syndromes combined with tinnitus, hearing impairment, vertigo, and nausea.	Type of study: not specified Sample size: 757 Tools: Multisensory neuro-otological tests	Results indicated that tinnitus is often part of a complex multisensory and central disorder, with a predominantly central rather than peripheral pathophysiological background.
(16) Valdete Alves Valentins dos Santos-Filha, Noise-induced tinnitus: auditory evoked potential in symptomatic and asymptomatic patients. Clinics (2014) [42]	PubMed	Evaluation of the central auditory pathways in workers with noise- induced tinnitus with normal hearing thresholds, compared the ABR results in groups with and without tinnitus and correlated the tinnitus location to the auditory brainstem response findings in individuals with a history of occupational noise exposure.	Type of study: not specified Sample size: 60 Tools: Audiometry, ABR	Although mean ABR latencies did not significantly differ, tinnitus patients showed more frequent qualitative abnormalities in lower brainstem responses, particularly in bilateral tinnitus cases.
(17) Louis Negri, Optimization of the Operant Silent Gap-in-Noise Detection Paradigm in Humans. J. Integr. Neurosci. (2024) [43]	PubMed	In the auditory domain, temporal resolution is the ability to respond to rapid changes in the envelope of a sound over time. Silent gap-in-noise detection tests assess temporal resolution. Whether temporal resolution is impaired in tinnitus and whether those tests are useful for identifying the condition is still debated. The aim is to revisit these questions by assessing the silent gap-in-noise detection performance of human participants.	Type of study: not specified Sample size: 71 Tools: Gap detection tasks, GPIAS, ABR	Tinnitus participants had higher gap detection thresholds at certain frequencies compared with controls. ABR latencies varied with tinnitus severity, suggesting impaired temporal resolution but limited individual diagnostic value.
(18) Raquel Mezzalira, The Contribution of Otoneurological Evaluation to Tinnitus Diagnosis. International Tinnitus Journal (2004) [44]	Scopus	The aim of this study is to analyze the contribution of otoneurological evaluation in the diagnosis of tinnitus.	Type of study: not specified Sample size: 195 Tools: Clinical history, audiometry, vestibular tests	Otoneurological evaluation contributed to reaching an etiological diagnosis in most cases, supporting its usefulness in routine tinnitus workup.
(19) Pauline Devolder, The role of hidden hearing loss in tinnitus: Insights from early markers of peripheral hearing damage. Hearing Research (2024) [45]	Scopus	This study investigated three potential markers of peripheral hidden hearing loss in subjects with tinnitus: extended high-frequency audiometric thresholds, the ABR, and the envelope following response	Type of study: not specified Sample size: 54 Tools: High-frequency audiometry, ABR, Envelope following response, speech-in-noise	No significant differences in markers of synaptopathy were found between tinnitus and controls. Older tinnitus patients performed better in low-pass filtered speech-in-noise tests, possibly linked to hyperacusis.
(20) Cornelis P. Lanting, Unilateral Tinnitus: Changes in Connectivity and Response Lateralization Measured with fMRI. PLoS ONE (2014) [46]	Scopus	Tinnitus is a percept of sound that is not related to an acoustic source outside the body. For many forms of tinnitus, mechanisms in the central nervous system are believed to play a role in the pathology. In this work were specifically assessed possible neural correlates of unilateral tinnitus.	Type of study: not specified Sample size: 30 (control group: 16, study group: 14) Tools: fMRI (auditory network connectivity)	Patients with tinnitus showed reduced lateralization of auditory responses and decreased connectivity between brainstem and cortex. The cerebellar vermis appeared to participate in tinnitus-related plasticity.

**Table 3 audiolres-15-00149-t003:** Quality assessment: EPHPP scores.

Author, Year [Ref. Num]	EPHPP Scores												
	SB		D		C		B		DC		DO		Overall
	R1	R2	R1	R2	R1	R2	R1	R2	R1	R2	R1	R2	
Saeid Mahmoudian, 2013 [27]	W	W	M	W	W	W	M	M	S	M	W	W	W
Haúla F. Haider, 2022 [28]	W	W	W	W	W	M	M	M	S	S	W	W	W
Sujoy Kumar Makar, 2017 [29]	W	W	M	W	W	M	S	S	W	W	M	M	W
Jae Ho Oh, 2013 [30]	M	W	W	W	W	W	W	W	M	W	M	M	W
Shadman Nemati, 2014 [31]	W	W	W	W	W	W	M	M	M	W	M	W	W
Helga M. Kehrle, 2008 [32]	S	M	M	S	W	W	W	W	M	W	M	M	W
Olaf Zagólski, 2016 [33]	M	M	W	M	W	W	M	W	W	M	n/a	n/a	W
Hyun Joon Shim, 2017 [34]	W	W	W	W	W	W	S	M	M	M	M	M	W
Aysun Coskunoglu, 2017 [35]	M	M	M	M	M	M	S	M	S	M	M	S	S
C.P. Lanting, 2008 [36]	W	W	W	M	W	W	M	M	M	W	M	M	W
Hsiang-Hung Lee, 2025 [37]	W	W	M	M	M	M	S	S	S	M	M	M	M
Eui-Cheol Nam, 2005 [38]	W	W	n/a	n/a	W	M	S	M	W	W	M	W	W
Bernhard Schick, 2001 [39]	M	M	W	M	W	W	W	W	M	W	M	W	W
Valdete Alves Valentins dos Santos Filha, 2015 [40]	M	M	W	M	W	W	M	M	M	M	W	W	W
Claus-F. Claussen, 2009 [41]	S	S	S	S	S	S	S	S	S	M	M	M	S
Valdete Alves Valentins dos Santos Filha, 2014 [42]	M	W	W	M	W	W	S	S	W	W	M	M	W
Louis Negri, 2024 [43]	M	M	M	M	S	S	S	S	S	S	M	M	S
Raquel Mezzalira, 2004 [44]	W	W	W	W	W	W	W	W	W	W	W	W	W
Pauline Devolder, 2024 [45]	W	W	W	W	W	W	M	M	S	S	M	M	W
Cornelis P. Lanting, 2014 [46]	W	W	W	M	W	W	S	S	S	M	M	W	W

SB = Selection Bias, D = Study Design, C = Confounders, B = Blinding, DC = Data Collection Method, DO = Withdrawals and Dropouts; W = Weak, M = Moderate, S = Strong, n/a = not applicable.

**Table 4 audiolres-15-00149-t004:** The radiological findings from the two cited studies are listed in order of decreasing incidence.

3D-FIESTA	MRI
1. Acoustic neuroma	1. Various degrees of subcortical and periventricular microangiopathic gliosis
2. EVAS	2. Pontine infarctions
3. PICA aneurysm	3. Loops of the inferior anterior cerebellar artery
4. Inner ear anomaly	4. Acoustic neuroma
5. Hypoplastic 8th n.	5. Inflammation of the eight cranial nerve
6. Vertebral a. calcification	6. Meningioma at the cerebellopontine angle
7. Pons infarction	7. Cerebellar infarctions
8. Epidermal cyst	8. Labyrinthine hemorrhage
	9. Pachymeningiosis affecting the internal auditory meatus
	10. Cerebral multiple sclerosis
	11. Cerebral venous dysplasia
	12. Parietal meningioma
	13. Cerebral atrophy with ventricular dilatation
	14. Cochlea enhancement
	15. Perilymphatic fistula
	16. Pontine multiple sclerosis
	17. Pontine venous dysplasia
	18. Cerebral sarcoidosis
	19. Cerebral cavernoma
	20. Temporal lobe mucoepidermoid metastasis
	21. Aqueduct stenosis
	22. Ventricular cyst
	23. Subdural hematoma
	24. Posttraumatic cerebral gliosis
	25. Cholesterin granuloma of the petrous bone apex
	26. Chondrosarcoma of the petrous bone apex
	27. Cystadenolymphoma of the parotid gland

## Data Availability

All information or data are included in this article.

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
