# Peer review of "An Insight into Role of Auditory Brainstem in Tinnitus: A Systematic Review of Diagnostic Assessments"

_audiolres, 2025, doi:10.3390/audiolres15060149_

Round 1
Reviewer 1 Report
Comments and Suggestions for Authors
Dear author(s),
I have reviewed the manuscript. My response is given in a point-by-point manner below.
Sincerely
An insight on the role of brainstem on tinnitus: A Systematic Review on the diagnostic tools
Title:
- I think “Auditory brainstem” is better than “brainstem” in the title.
- Also, you can change “diagnostic tools” to the “diagnostic assessments”.
Abstract:
- As the multi-origin nature of tinnitus, what was your reason for investigating merely the “brainstem” in this paper?
- Wouldn't it have been better to examine the cerebral cortex and even non-auditory areas, such as the role of the frontal cortex, in tinnitus?
- In Methods section, you mention “otoacoustic emissions”. As the DPOAE and TEOAE investigate OHC of the cochlea, what is your reason to apply them in the current paper?
- In keyword section, remove “ABR wave V”. Also, add “Auditory Brainstem Response”.
Main Text:
- Introduction
- In 2nd paragraph, use “inferior colliculus (IC)” instead of “IC”.
- Since the multi-origin nature of tinnitus, what was your reason for investigating merely the “brainstem” in this study?
- Since the OAE investigate OHC of the cochlea, what is your reason to apply it in the current paper?
- As “middle- and late-latency evoked potentials (AMLR & ALLR)” investigate mainly “Auditory Cortex”, what is the reason for apply them in this study?
- Materials and Method
2.1. Inclusion Criteria and Eligibility
- The “Eligible studies” (Intervention section) is ambiguous, clear it more.
2.2. Search Strategy
2.3. Study Selection and Data Extraction
- Both above sections are good.
- Results
- Figure 1 is excellent.
- Table 1 is very space-consuming; it spans 12 pages! Revise and modify it
3.1. The Quality Assessment
- This section is suitable.
- Discussion
- In 2nd paragraph, use “Inter-peak Latency (IPL)” instead of “IPL”.
- Add the “Limitations of current study” to end of the Discussion section.
- Conclusions
- I think the Conclusions section is OK. (The only suggestion to authors is that “move the suggestion for future studies in to the Discussion section”).
Author Response
REVIEWER 1
I have reviewed the manuscript. My response is given in a point-by-point manner below.
Sincerely
An insight on the role of brainstem on tinnitus: A Systematic Review on the diagnostic tools
Title:
- I think “Auditory brainstem” is better than “brainstem” in the title.
[Reply]: Thank you for your suggestion. We have modified the text accordingly
- Also, you can change “diagnostic tools” to the “diagnostic assessments”.
[Reply]: Thank you for your suggestion. We have modified the text accordingly
Abstract:
- As the multi-origin nature of tinnitus, what was your reason for investigating merely the “brainstem” in this paper?
[Reply]: Thank you for this question. We focused on the auditory brainstem because it is the first central relay for auditory signals, a key site of abnormal synchrony and central gain, and is increasingly recognized as crucial in tinnitus generation and modulation even in patients with normal hearing. We have now clarified this issue in the abstract (lines: 26-30).
- Wouldn't it have been better to examine the cerebral cortex and even non-auditory areas, such as the role of the frontal cortex, in tinnitus?
[Reply]: Thank you for this question. We agree that cortical and non-auditory areas contribute to tinnitus; however, the present review focuses on brainstem-related diagnostic findings.
This point has been included in the limitation section at the end of the discussion.
- In Methods section, you mention “otoacoustic emissions”. As the DPOAE and TEOAE investigate OHC of the cochlea, what is your reason to apply them in the current paper?
[Reply]: Thank you for your question. Although OAEs mainly assess outer hair cell function, they are routinely used to exclude peripheral cochlear dysfunction when evaluating central auditory abnormalities, including brainstem involvement. We have now clarified this point in the Abstract at line 39.
- In keyword section, remove “ABR wave V”. Also, add “Auditory Brainstem Response”.
[Reply]: Thank you for your suggestion. We have modified the keyword section accordingly.
Main Text:
- Introduction
- In 2nd paragraph, use “inferior colliculus (IC)” instead of “IC”.
[Reply]: Thank you for your suggestion. We have modified the text accordingly
- Since the multi-origin nature of tinnitus, what was your reason for investigating merely the “brainstem” in this study?
[Reply]: Thank you for your question. We stated about this choice in Introduction section (lines: 117-122).
- Since the OAE investigate OHC of the cochlea, what is your reason to apply it in the current paper?
[Reply]: Thanks: OAE are routinely used to exclude peripheral cochlear dysfunction when evaluating central auditory abnormalities, including brainstem involvement.
We added a new sentence in the Introduction section to better explain our choice (lines: 108-113)
- As “middle- and late-latency evoked potentials (AMLR & ALLR)” investigate mainly “Auditory Cortex”, what is the reason for apply them in this study?
[Reply]: Thank you for this question. We included middle- and late-latency responses because, despite partially reflecting cortical activity, they could also define retrocochlear or brainstem-related abnormalities.
- Materials and Method
2.1. Inclusion Criteria and Eligibility
- The “Eligible studies” (Intervention section) is ambiguous, clear it more.
[Reply]: Thanks. We included observational (case–control, cohort) and RCTs studies and excluded case reports, small case series, and non–peer-reviewed papers. This point has been clarified at lines: 140-142
2.2. Search Strategy
2.3. Study Selection and Data Extraction
- Both above sections are good.
[Reply]: Thank you for your positive comment.
- Results
- Figure 1 is excellent.
[Reply]: Thank you for your positive comment.
- Table 2 is very space-consuming; it spans 12 pages! Revise and modify it
[Reply]: Thank you for your suggestion. We have summarized Table 2 as much as possible, clarified abbreviations in the caption, and added details on tinnitus type and control groups, while unnecessary p-values have been removed.
3.1. The Quality Assessment
- This section is suitable.
[Reply]: Thank you for your positive comment.
- Discussion
- In 2nd paragraph, use “Inter-peak Latency (IPL)” instead of “IPL”.
[Reply]: Thank you for your suggestion. We have modified the text accordingly removing IPL at line 215, and Table 2 we used the extended form.
- Add the “Limitations of current study” to end of the Discussion section.
[Reply]: Thank you for your suggestion: we have added a dedicated “Limitations of the current study” paragraph at the end of the discussion section, as requested.
- Conclusions
- I think the Conclusions section is OK. (The only suggestion to authors is that “move the suggestion for future studies in to the Discussion section”).
[Reply]: Thank you for your suggestion. We have modified the text accordingly.
Reviewer 2 Report
Comments and Suggestions for Authors
this paper is well written.
Author Response
REVIEWER 2
this paper is well written.
[Reply]: Thank you for your positive comment.
Reviewer 3 Report
Comments and Suggestions for Authors
I would like to start by thanking you for the opportunity to review this manuscript. This review examines the role of the brainstem in tinnitus, an important topic since tinnitus is not solely an issue related to the inner ear. The subject of this manuscript is intriguing; however, several issues need to be addressed before it can be considered for publication. My specific recommendations are outlined below. It is crucial to pay special attention to the inclusion criteria for the articles in the review. While the authors state that they considered case-control and cohort studies, not all included articles fully meet these criteria. There are two potential approaches: either include articles more strictly based on the current criteria or expand the criteria. If the latter is chosen, additional articles should also be considered.
Introduction
To introduce tinnitus, it is essential to explain its main types: objective or subjective, and primary and secondary. Additionally, the primary causes of secondary cases should be noted. Regarding this, please cite the following reference article:
Lines 52-53. The concern regarding tinnitus in individuals with normal hearing is further supported by the fact that tinnitus can impair speech understanding, even if hearing thresholds are normal. Regarding this, please cite the following reference article:
doi: 10.3389/fneur.2025.1672762.
Line 99. While otoacoustic emission is important is important in differentiating between central and peripheral auditory problems, its main mechanism is to analyse the functioning of the outer hair cells; this needs clarification.
Materials and Methods
Table 1. Please do not write ‘or’ in capitals as it generally refers to the odds ratio.
Results
Table 1. Each abbreviation used in the table should be clearly explained in the table caption. Based on Haúla F. Haider's study, "pure-tone audiometry" would be a more precise term instead of simply "audiometry." Sujoy Kumar Maka’s study suggests that "high-frequency sensorineural hearing loss" is a more accurate choice. Overall, for each study discussed, it would be beneficial to specify the characteristics of the tinnitus group examined, including whether they are acute or chronic, primary or secondary cases of tinnitus, and if they are unilateral or bilateral. Additionally, details about the control group selection should be included. I do not recommend including p-values from previous investigations; stating that there is a statistically significant difference is sufficient. Without mentioning the statistical tests used or the significance levels, including p-values does not add value. When studies involve a control group, it would be useful to know if the authors indicated that the control group is statistically comparable to the study group. The phrase "One-way analysis of variance and Chi-squared tests were applied" is unnecessary, or if included, the statistical tests used for each article should be consistent. Another issue is that in some instances, the study type is not specified, or the distribution of cases and controls is missing. However, the inclusion criteria state that randomized-controlled and cohort studies were considered. In this case, more recently published studies should also be included. In Bernhard Schick's study, no controls were mentioned. If this study is to be included, the inclusion criteria need to be updated. However, in this case, more articles addressing the same topic should be considered. Additionally, the statement "Thus, we analysed the central auditory pathways..." is inappropriate since these are not the authors' results, but rather a discussion of previous results. The term "anamnesis" should be replaced with "case history," as it is more appropriate. Finally, the phrase "Often, we find multisensory syndromes..." should be revised to avoid the use of "we" in reference to the authors, as this follows the same concern mentioned earlier.
Quality assessment was not addressed in the methods section. The methods section should clearly outline each method used in the results.
Discussion
Lines 233-235. It is unnecessary to restate the results of a previous investigation in the discussion.
Table 4. Acoustic neuroma should not be classified as an inner ear disorder because it is primarily located in the internal auditory canal, not the inner ear. Additionally, each abbreviation used in this table should be clearly defined in the table caption.
The Funding section needs to be corrected.
I am looking forward to receiving the revised version of the manuscript, which includes a point-by-point response to each review comment with all required changes accurately made. This is necessary for deciding whether this manuscript can be considered for publication.
Author Response
REVIEWER 3
I would like to start by thanking you for the opportunity to review this manuscript. This review examines the role of the brainstem in tinnitus, an important topic since tinnitus is not solely an issue related to the inner ear. The subject of this manuscript is intriguing; however, several issues need to be addressed before it can be considered for publication. My specific recommendations are outlined below. It is crucial to pay special attention to the inclusion criteria for the articles in the review. While the authors state that they considered case-control and cohort studies, not all included articles fully meet these criteria. There are two potential approaches: either include articles more strictly based on the current criteria or expand the criteria. If the latter is chosen, additional articles should also be considered.
Introduction
- To introduce tinnitus, it is essential to explain its main types: objective or subjective, and primary and secondary.
[Reply]: Thanks for you for your suggestion. We have now implemented the introduction section accordingly (lines 53-57).
- Additionally, the primary causes of secondary cases should be noted. Regarding this, please cite the following reference article:
Lines 52-53. The concern regarding tinnitus in individuals with normal hearing is further supported by the fact that tinnitus can impair speech understanding, even if hearing thresholds are normal. Regarding this, please cite the following reference article:
doi: 10.3389/fneur.2025.1672762.
[Reply]: Thank you for your suggestion. The reference has been included at line 62.
- Line 99. While otoacoustic emission is important in differentiating between central and peripheral auditory problems, its main mechanism is to analyse the functioning of the outer hair cells; this needs clarification.
[Reply]: Thank you for your question. This issue has been clearified now in the introduction section, lines: 108-113
Materials and Methods
- Table 1. Please do not write ‘or’ in capitals as it generally refers to the odds ratio.
[Reply]: Thank you for your suggestion. In Table 1 we have reported the code of the search string generated by the platforms based on the entered keywords, thus avoiding confusion with odds ratio.
Results
- Table 2. Each abbreviation used in the table should be clearly explained in the table caption. Based on Haúla F. Haider's study, "pure-tone audiometry" would be a more precise term instead of simply "audiometry." Sujoy Kumar Maka’s study suggests that "high-frequency sensorineural hearing loss" is a more accurate choice. Overall, for each study discussed, it would be beneficial to specify the characteristics of the tinnitus group examined, including whether they are acute or chronic, primary or secondary cases of tinnitus, and if they are unilateral or bilateral. Additionally, details about the control group selection should be included. I do not recommend including p-values from previous investigations; stating that there is a statistically significant difference is sufficient. Without mentioning the statistical tests used or the significance levels, including p-values does not add value. When studies involve a control group, it would be useful to know if the authors indicated that the control group is statistically comparable to the study group. The phrase "One-way analysis of variance and Chi-squared tests were applied" is unnecessary, or if included, the statistical tests used for each article should be consistent. Another issue is that in some instances, the study type is not specified, or the distribution of cases and controls is missing. However, the inclusion criteria state that randomized-controlled and cohort studies were considered. In this case, more recently published studies should also be included. In Bernhard Schick's study, no controls were mentioned. If this study is to be included, the inclusion criteria need to be updated. However, in this case, more articles addressing the same topic should be considered. Additionally, the statement "Thus, we analysed the central auditory pathways..." is inappropriate since these are not the authors' results, but rather a discussion of previous results. The term "anamnesis" should be replaced with "case history," as it is more appropriate. Finally, the phrase "Often, we find multisensory syndromes..." should be revised to avoid the use of "we" in reference to the authors, as this follows the same concern mentioned earlier.
[Reply]: Thank you for your suggestions. We have summarized Table 2 as much as possible, clarified abbreviations in the caption; also, we have added details on tinnitus type and control groups, while unnecessary p-values were removed.
Please note that we have further revised the table based on comments from other reviewers.
- Quality assessment was not addressed in the methods section. The methods section should clearly outline each method used in the results.
[Reply]: Thank you for your suggestions: we have included a ‘Quality Assessment’ section in the methods section as suggested. Particularly, we have divided the results subsection, "The Quality assessment" in two paragraphs (subsection 2.4 and 3.1)
Discussion
- Lines 233-235. It is unnecessary to restate the results of a previous investigation in the discussion.
[Reply]: Thanks for your suggestion; however, we would like to provide just a brief description of the cited papers to be more accurate in the discussion. We are confident that the Reviewer will agree with this choice.
- Table 4. Acoustic neuroma should not be classified as an inner ear disorder because it is primarily located in the internal auditory canal, not the inner ear. Additionally, each abbreviation used in this table should be clearly defined in the table caption.
[Reply]: Thanks for your comment. We agree that acoustic neuromas are not inner ear disorders, however, we have reported in table 4 the radiological findings of the two cited studies. Table caption has been implemented accordingly.
- The Funding section needs to be corrected.
[Reply]: Thank you for your suggestion. We have implemented the section accordingly
I am looking forward to receiving the revised version of the manuscript, which includes a point-by-point response to each review comment with all required changes accurately made. This is necessary for deciding whether this manuscript can be considered for publication.
Reviewer 4 Report
Comments and Suggestions for Authorsㅍ
Methods section
- Inclusion/exclusion criteria are clear, but the research question (“diagnostic of brainstem pathology that generate the tinnitus”) needs refinement in grammatical clarity (suggest: “What diagnostic tools can identify brainstem pathology contributing to tinnitus?”).
- Eligibility criteria list only RCTs and cohort studies, but Table 2 includes case-control and descriptive studies. This inconsistency needs clarification.
- OSF registration is commendable, but ensure the DOI link is active and accessible.
Results section
- Tables are comprehensive but could benefit from standardized presentation (e.g., consistent reporting of ABR wave abnormalities, imaging findings, and outcome measures).
- Statistical findings from included studies (effect sizes, p-values) are sometimes summarized qualitatively. Adding quantitative synthesis where possible (e.g., pooled latency differences in ABR) would enhance rigor.
- The PRISMA flow diagram is included, but figure clarity should be improved for publication (higher resolution, clear labels).
Author Response
REVIEWER 4
Methods section
- Inclusion/exclusion criteria are clear, but the research question (“diagnostic of brainstem pathology that generate the tinnitus”) needs refinement in grammatical clarity (suggest: “What diagnostic tools can identify brainstem pathology contributing to tinnitus?”).
[Reply]: Thank you for your suggestion, we have rephrased as suggested.
- Eligibility criteria list only RCTs and cohort studies, but Table 2 includes case-control and descriptive studies. This inconsistency needs clarification.
[Reply]: Thank you for your question. We revised the Eligibility criteria list and explicitly included observational studies.
- OSF registration is commendable, but ensure the DOI link is active and accessible.
[Reply]: Thank you for your question. We have checked and confirm that the doi code is correct. https://doi.org/10.17605/OSF.IO/UNXYV
Results section
- Tables are comprehensive but could benefit from standardized presentation (e.g., consistent reporting of ABR wave abnormalities, imaging findings, and outcome measures).
[Reply]: Thanks for your question. Please note that the tables have now been rearranged based on the recommendations of the other three reviewers.
- Statistical findings from included studies (effect sizes, p-values) are sometimes summarized qualitatively. Adding quantitative synthesis where possible (e.g., pooled latency differences in ABR) would enhance rigor.
[Reply]: Thanks for your question. However, since this specific data was not available from all selected documents, it was not possible to include this information.
- The PRISMA flow diagram is included, but figure clarity should be improved for publication (higher resolution, clear labels).
[Reply]: Thanks for your suggestion, we improved the figure as requested.
Round 2
Reviewer 1 Report
Comments and Suggestions for Authors
Thank the authors of the manuscript for implementing all the comments I had made.
In my opinion, no further amendments are necessary.
Good luck!
Author Response
The authors thank Reviewer 1 for his constructive comments.
Sincerely,
Prof. Anna Rita Fetoni and dr. Nicola Serra
Reviewer 3 Report
Comments and Suggestions for Authors
Thank you for sending me the revised version of the manuscript. The authors have made significant efforts to enhance its quality. Although not all recommended corrections were adopted, the most critical ones have been effectively addressed. Therefore, in my opinion, this manuscript can now be considered for publication.
Author Response
The authors thank Reviewer 3 for his constructive comments.
Sincerely,
Prof. Anna Rita Fetoni and dr. Nicola Serra